# Microwave Control of *Reynoutria japonica* Houtt., Including Ecotoxicological Aspects and the Resveratrol Content in Rhizomes

**DOI:** 10.3390/plants13020152

**Published:** 2024-01-05

**Authors:** Krzysztof Słowiński, Beata Grygierzec, Agnieszka Baran, Sylwester Tabor, Diletta Piatti, Filippo Maggi, Agnieszka Synowiec

**Affiliations:** 1Department of Forest Management, Engineering and Technology, University of Agriculture in Krakow, al. 29 Listopada 46, 31-425 Krakow, Poland; krzyszof.slowinski@urk.edu.pl; 2Department of Agroecology and Crop Production, University of Agriculture in Krakow, al. Mickiewicza 21, 31-120 Krakow, Poland; beata.grygierzec@urk.edu.pl; 3Department of Agricultural and Environmental Chemistry, University of Agriculture in Krakow, al. Mickiewicza 21, 31-120 Krakow, Poland; agnieszka.baran@urk.edu.pl; 4Department of Production Engineering, Logistics and Applied Computer Science, University of Agriculture in Krakow, ul. Balicka 116 B, 30-149 Krakow, Poland; sylwester.tabor@urk.edu.pl; 5School of Pharmacy, University of Camerino, Via Sant’Agostino 1, 62032 Camerino, Italy; diletta.piatti@unicam.it (D.P.); filippo.maggi@unicam.it (F.M.)

**Keywords:** electromagnetic microwaves, invasive weed, non-chemical weed control, physical control, HPLC-MS/MS

## Abstract

Japanese knotweed (*Reynoutria japonica* Houtt.) is Poland’s invasive weed, for which there is no efficient control method. The rhizomes of this species are rich in resveratrol. In this work, we evaluated (1) the effectiveness of electromagnetic microwaves (MV) in destroying Japanese knotweed using an original device, HOGWEED (MV of 2450 MHz), (2) the ecotoxic effect of the MV on the soil environment, and (3) the resveratrol content in knotweed rhizomes after MV treatment. The field studies were carried out in 2022 in southern Poland. Cut plants were MV-treated for times of 5.0–25.0 min. The MV efficiency was checked 10 and 56 days after treatment (DAT). After MV treatment, fresh soil samples were taken to analyze their ecotoxicity. As a result, at 56 DAT, knotweed was controlled if MV was used for at least 20.0 min. The MV did not affect the soil ecotoxicity. The MV-treated soils were classified as non-toxic or low-toxic. To analyze the resveratrol content, healthy knotweed rhizomes were dug out, treated with MV in the laboratory at 2.5–10.0 min, and analyzed for resveratrol content in HPLC-MS/MS. As a result, the resveratrol in the rhizomes significantly decreased in a time-dependent manner following MV exposure.

## 1. Introduction

There are six globally recognized species within the *Reynoutria* Houtt. genus, also known as *Polygonum* Sieb. and Zucc. and *Fallopia* Houtt. These species are native to eastern Asia, spanning from the eastern part of Russia to the Indochina Peninsula [1]. However, they have been introduced to Europe and North America, where they have become widespread and are designated as highly invasive alien species (IAS). As a result, they are included on Switzerland’s blacklist of IAS, indicating their status as one of the most harmful environmental weeds [2]. In Poland, Japanese knotweed (*Reynoutria japonica* Houtt., also known as *Fallopia japonica* (Houtt.) Ronse Decr.), Sakhalin knotweed (*Reynoutria sachalinensis* F. Schmidt), and a hybrid of the two called Czech knotweed (*Reynoutria × bohemica*) have been identified [3,4]. All three of these species are considered extremely hazardous to the native flora in Poland [5].

Japanese knotweed is the most widespread of these invasive knotweeds in Poland [5], and it is also recognized as one of the top 100 most invasive species globally [6]. Remarkably, a new plant can sprout from a rhizome fragment as small as 1 cm long and weigh less than 0.7 g when placed in soil or water or from a small shoot section containing a single node [7,8]. The species’ invasion is facilitated by its substantial size, which casts shade on other plants, its high regenerative potential, and its allelopathic effects [9,10]. The spread of the plant is aided by the dispersion of rhizomes via water, such as during river floods, contributing to the spread of these plants along river valleys [4,7,11]. Japanese knotweed can also be found in wastelands, fallow fields, roadsides, railway embankments, and agricultural areas. Consequently, the dense thickets formed by Japanese knotweed reduce the biodiversity of natural and semi-natural habitats [4].

However, it is important to note that Japanese knotweed also possesses numerous valuable biologically active compounds [12] and is consumed as an edible plant in Japan [13]. Its rhizomes are a source of resveratrol, a polyphenolic phytoalexin (3,5,4′-trihydroxy-*trans*-stilbene) belonging to the stilbene group [14]. Resveratrol has been shown to have benefits for various health conditions, including cardiovascular, inflammatory, neurodegenerative, metabolic, and age-related diseases [15,16]. The resveratrol content in Japanese knotweed rhizomes is reported to be approximately 7.4 to 11.1 times higher than that in their leaves or stems, ranging from 2.96 to 3.77 mg/g of dry material [17,18].

Managing invasive knotweeds is challenging due to their ability to accumulate significant below-ground reserves, allowing them to recover from various control methods, including physical, biological, chemical, and integrated approaches [19,20,21,22]. In the case of Japanese knotweed, some success has been achieved by focusing on seasonal resource translocation between above- and below-ground biomass and using suitable herbicide coverage or injections, such as glyphosate [19,20,21,22]. Another method involves physically burying rhizomes and stems [23]. Nevertheless, these methods have proven ineffective, prompting the search for new approaches to control Japanese knotweed [24].

Microwave radiation has garnered interest in environmental studies [25,26,27] and has been employed to control giant hogweed populations [28]. However, the impact of radiation on soil biological activity remains unclear. Some research suggests that microwave radiation applied to soil may adversely affect its physicochemical and biological properties [27]. Conversely, other studies have indicated that microwave soil treatment does not significantly alter soil pH or nutrient availability. Still, it can reduce the population of soil-colonizing bacteria as irradiation time increases [25,29].

This research aimed to achieve two objectives: (1) evaluate the efficacy of microwave radiation in eradicating Japanese knotweed plants while considering its ecotoxic effects on the soil environment and (2) analyze the resveratrol content in knotweed rhizomes following microwave treatment.

## 2. Results

### 2.1. Efficacy of Japanese Knotweed Control Using Microwaves

One day before the MV treatment, there was a precipitation of 13 mm in the studied area (Figure 1), which created good conditions for the treatment, as the soil was moist (88%). The average temperature of the control plants, i.e., untreated plants and the surrounding soil, was 21.9 °C. The temperature of MV-treated plants for 5.0, 7.5, 10.0, 15.0, 20.0, or 25.0 min reached 45.4, 53.7, 67.0, 77.6, 84.8, and 89.6 °C, respectively (Table 1).

The polynomial correlation fit very well, as the determination index was almost equal to 1 (Figure 2). The temperature of the cut shoots of Japanese knotweed and the surrounding soil surface increased with the increase in the duration of the duration of MV irradiation. The temperature increased between the MV for 10 and 25 min by 22.6 °C, and for between 20 and 25 min, it increased only by 4.8 °C.

Ten days after MV, the controlling effect of MV depended on the treatment time. Plants MV-treated for 5.0, 7.5, 10.0, or 15.0 min regenerated, and 1–2 cm long new shoots appeared. On the contrary, plants MV-treated for 20.0 or 25.0 min did not regrow. During this period, there was precipitation in the study site almost every day, a total of 35 mm, and the average air temperature was ca. 19 °C (Figure 1).

The photos below present selected radiation plots 10 days after the MV treatment (Figure 3A–C).

Fifty-six days after the MV treatment, no regrowth of Japanese knotweed was observed on plots where the MV was used for 20.0 or 25.0 min (Figure 4A,B).

### 2.2. Soil Ecotoxicity after Microwave Radiation

Soil ecotoxicity results are presented in Table 2. In the Phytotoxkit test, *Sinapis alba* L. germination inhibition ranged from 0 to 21%. The highest germination inhibition was found in the soils treated with MV for 7.5 and 20.0 min. Inhibition of *S. alba* root growth ranged from −10 to 43%. The greatest inhibition of root growth was shown in the soils MV-treated for 20.0 min, and the lowest inhibition was shown in those treated for 10.0 min. In soils treated for 10.0 min, root growth of *S. alba* was stimulated. The analyzed soil samples treated for 10.0 min were non-toxic to *S. alba*, and the remaining samples showed low toxicity. The germination inhibition of *Sorghum saccharatum* (L.) Moench was 0–8%. In general, 100% germination of the test plants was observed in most soil samples (except for the 7.5 min). The inhibition percentage of *S. saccharatum* root growth ranged from −25 to 5%. *S. saccharatum* root growth was stimulated in all the MV-treated soil, especially for 10.0 min. No significant effect of soil MV treatment on their toxicity for the test plants was found.

In the Ostracodtoxkit test, growth inhibition of the crustacean *Heterocypris incongruens* (Ramdohr 1808) ranged from 13 to 48%, and mortality was zero. The greatest inhibition of crustacean growth was demonstrated in MV soil treated for 25.0 min and in control soil (0 min). Notably, in all the MV-treated soils, a tendency to reduce ecotoxicity for crustaceans was observed, and the lowest inhibition was shown in the soil treated for 5.0 min. Soil treated for 5.0 min was non-toxic to crustaceans. The other MV-treated soils were of low toxicity.

Luminescence inhibition of *Aliivibrio fischeri* (Beijerinck 1889) in the MV-treated soils ranged from −9 to 26%. Notably, a statistically significant reduction in soil toxicity was found in soils treated for 5.0, 7.5, and 10.0 min. In soils treated for 15.0, 20.0, and 25.0 min, soil toxicity significantly increased for bacteria compared to the control (not-MV) soil. Luminescence stimulation was demonstrated in soils MV-treated for 5.0, 7.5, and 10.0 min. The remaining soils exposed to MV demonstrated a low-toxic effect for bacteria.

The influence of MV on soils’ increased ecotoxicity has not been demonstrated. According to the hazard classification, considering the responses of all soil organisms, soils MV-treated for 5.0 min were classified as the first toxicity class, meaning that they are non-toxic samples that do not threaten the environment. The remaining soils were classified as class II, meaning they are low-toxic and do not threaten the environment.

### 2.3. Content of Resveratrol in the Rhizomes of Japanese Knotweed Treated with Microwaves

The duration of MV treatment affected the temperature of the exposed, bare knotweed rhizomes in the laboratory, as was read from the thermal camera readings. The temperature of rhizomes increased several times following the shortest MV time. The average temperature of unexposed (control) rhizomes was 20 °C, while the rhizomes MV-treated for 2.5 min increased almost 3.5 times (Table 3). The temperature of the rhizomes MV-treated for subsequent periods also increased, compared to the 2.5 min MV time, but not so intensely: by 9%, 15%, and 27% for the 5.0, 7.5, and 10.0 min of MV irradiation. 

Following the MV treatments, the changes in rhizome temperature fit into a polynomial curve that presented a regression equation with a high determination index of 0.92 (Figure 5). 

As a result of the rhizomes MV treatment and the accompanying temperature rises, rhizomes’ weight loss was observed, most probably due to water evaporation. The highest weight loss was noted after MV for 10.0 min, by 29%, compared to the non-MV control. In addition, the following losses in rhizome mass following MV were recorded: 21% (7.5 min), 13% (5.0 min), and 6% (2.5 min), compared to non-MV control rhizomes.

The resveratrol content in control (non-MV) rhizomes averaged 164.1 µg/g (Figure 6). The MV treatment for 2.5 min resulted in a significant 63% decrease in resveratrol content. In the rhizomes MV-treated for 5.0 or 10.0 min, there was a further significant decrease in resveratrol content by an average of 94% compared to the control and 83% compared to the rhizomes MV-treated for 2.5 min.

## 3. Discussion

The results presented in this study concerning the control of Japanese knotweed using microwave radiation (MV) at 2450 MHz and a power of 800 watts are pioneering. It was found that the longer the exposure time of cut knotweed plants to MV, the lower their regeneration. It was effective to use longer time ranges of MV, i.e., 20 or 25 min, which generated a plant and soil surface temperature of about 85–90 °C. Consequently, the plants did not regrow 56 days after the MV was applied. Such a long time needed to destroy knotweed plants is most likely due to the ability of plants to regenerate from a deep system of rhizomes and roots that reach up to two meters deep into the soil [30]. So far, the MV control technique has been used against another invasive species—Sosnowski hogweed (*Heracleum Sosnowskyi* Manden.)—also after the prior cutting of aboveground biomass and exposure of plants to MV with a frequency of 2.45 GHz and a power density of 30.0 kW/m^2^ emitted by the tubular antenna [28]. In the case of Sosnowski hogweed, 10 min of MV was enough to destroy the plants and effectively prevent their further regrowth. Previous research on the impact of MV on plants shows that different species show different sensitivity to MV frequency, exposure time, and intensity of MV irradiation. For example, a stimulating effect was observed with low-power MV irradiation of sequoia (*Sequoia* Endl.) in vitro culture [31]. However, in the case of alfalfa (*Medicago sativa* L.), no differences in growth and biomass accumulation were found between plants MV irradiated at a frequency of 2.45 GHz and a power density of 0.5–1.2 mW/cm^2^ and plants that were not subjected to such treatment [32]. Senavirathna and Asaeda [33] also showed no effect of MV on the growth parameters of *Myriophyllum aquaticum* (Vell.) Verdc., which was exposed to horizontally and vertically polarized continuous MV with a frequency of 2 GHz and a power density of 1.8 W/m. Significant morphological modifications of the common bean (*Phaseolus vulgaris* L.) were caused by long-term MV with a frequency of 915 MHz and a maximum power density of 10 mW/m^2^ [34].

Apart from biological factors, environmental factors, particularly soil moisture, are important to guarantee the high effectiveness of destroying MV-treated plants. A study by Diprose et al. [35] shows that the higher the soil moisture when treating plants, the higher the effectiveness of the MV; indeed high soil moisture guarantees a high degree of moisture in plant tissues and a significant increase in their temperature. In our research, the shoots’ temperature and the substrate’s surrounding surface increased to 85–90 °C after 20.0–25.0 min of MV exposure. At such a high temperature, irreversible biochemical changes occur in plant tissues, i.e., denaturation of proteins and changes in their composition, changes in the composition of carbohydrates, and significant changes in the metabolism of amino acids and the glyoxylate and galactose pathways, which was observed after the use of MV in tissues of Sosnowski hogweed [28].

The ambiguous impact of MV on the soil environment [25,29] contributed to the study of its ecotoxicity to soil organisms. Our research used a battery of biotests to assess the ecological risk associated with the microwave method. The organisms used belonged to different taxonomic groups, representing different links in the trophic chain (producers, consumers, decomposers). They were characterized by different sensitivity to the conditions in the tested soil. It is worth noting that each species shows different sensitivity, and none are sensitive to all factors in the sample. In addition, biotests assess the total toxicity of all factors (and substances) present in the tested sample, often acting synergistically or antagonistically [36,37]. Such an approach in research assessing the risk associated with using MV is crucial [38]. In this study, in the context of the environmental effects of the presented MV method, their negligible side effect on the ecotoxicity of soils should be considered an important and promising result. According to the hazard classification, considering the responses of all soil organisms, soils MV-treated for 5.0 min were classified as samples that did not threaten the environment. The remaining soils were classified as class II, meaning they are low-toxic and do not threaten the environment.

Our studies showed a significant loss of resveratrol content in knotweed rhizomes directly MV-treated in the laboratory, already in the minimum time range of 2.5 min. The content of this compound in the tissues was reduced by more than 60% compared to the control rhizomes. Plants MV-treated for 5.0 or 10.0 min had 90% lower resveratrol concentration than the control. At the same time, there were no significant differences in the content of this compound in rhizomes irradiated for 5.0 and 10.0 min. The high temperature during the MV treatment most likely caused the knotweed cells and tissues to be degraded. That could have impacted, among other factors, the amount of resveratrol. According to Korablev et al. [39], at high temperatures, there is hyperfluidity of plant tissue membranes, leading to the outflow of ions, further reducing the activity of many enzymes. A significant reduction in resveratrol in Japanese knotweed rhizomes can also be explained by the degradation of polyphenolic compounds under the influence of high temperatures. However, the lack of significant differentiation of the content of this compound with longer MV exposure was probably the result of lower tissue moisture. The lower the tissue’s water content, the lower the degradation of polyphenols. Water molecules quickly absorb MV energy and generate heat, so water molecules move faster than other solvents [40]. In studies by Yuan et al. [41], the exposure to MV of 500 W for 8 min and a temperature of 60 °C increased the resveratrol and gallic acid degradation with an increase in the water percentage. Resveratrol and gallic acid were oxidized by free radicals formed by weak hydrogen bond breaks of molecular dipoles with increasing solvent temperature. In turn, the higher the percentage of water, the faster the solvent molecules moved. Physical stresses, such as UV-C radiation, may increase resveratrol content in knotweed tissues, as observed in Japanese knotweed [42]. Under the influence of UV-C radiation for 10 min, followed by incubation in the dark, the content of resveratrol increased by 2.6-fold and 1.6-fold, respectively, in UV-treated leaves. Differential expression analysis showed that Japanese knotweed strongly induced genes directly involved in resveratrol synthesis under UV-C.

## 4. Materials and Methods

### 4.1. Characteristics of the Study Site and Japanese Knotweed Population

Field studies were carried out in 2022 on a population of Japanese knotweed (*Reynoutria japonica* Houtt.) of approx. 250 m^2^ area located near the Vistula embankments in Kraków, south of Poland (N: 50°02′34.31″ E: 19°52′43.31″, Figure 7A,B). The plant material was collected respecting the IUCN policy [43], and required permission for collecting plant samples was obtained from the Municipal Greenery Board in Kraków, PL. The specimens were botanically identified by Dr. B. Grygierzec, based on [3], then archived in the herbarium resources of the Department of Agroecology and Crop Production of the University of Agriculture in Krakow, which provides access to deposited material.

#### Chemical Characteristics of Soil and Weather Data from the Study Site

The soil samples taken from the site were air-dried and sieved through a sieve with a mesh size of 2 mm. The following parameters were determined: granulometric composition by the areometric method; pH in 1 mol KCl dm^−3^ by the potentiometric method; and total organic carbon, nitrogen, and sulfur using the CNS Elementar apparatus Vario MAX cube (Elementar Analysensysteme GmbH, Langenselbold, Germany). Moreover, the content of assimilable phosphorus and potassium by the Egner–Riehm method was analyzed. The total content of elements (Ca, Mg, K, P, Na, Zn, Cd, Pb, Cu, Ni, Cr, Fe, Mn) in the soil was determined after hot mineralization in a mixture of HNO_3_ and HClO_3_ acids (3:2 *v*/*v*) (supra pure, MERCK, Rahway, NJ, USA) by ICP-OES (Inductively Coupled Plasma Atomic Emission Spectroscopy) on Optima 7300 DV (Perkin-Elmer, Waltham, MA, USA). Table 4 presents the characteristics of soil in the study site.

Weather data, i.e., daily temperatures and precipitation, recorded every hour from 28 June to 27 August 2022, were collected from the weather station of the Institute of Meteorology and Water Management in Kraków-Balice, located ca. 7.5 km from the study site. 

### 4.2. Control of Japanese Knotweed Using Microwaves

On 29 June 2022, the aboveground plant biomass was cut and removed from the ground selected for microwave radiation (MV). The cut shoots remaining in the soil were treated with a self-propelled device, HOGWEED, on a tracked chassis with a rotary mower intended to destroy invasive plants (Figure 8). The HOGWEED comprises an MV emitter consisting of nine magnetrons generating electromagnetic waves at 2450 MHz, each with a power of 800 W. The average surface power density for the dimensions of the antenna is 30 kW/m^2^. 

The MV with an open beam directed at the plants was used at 5.0, 7.5, 10.0, 15.0, 20.0, and 25.0 min in 9 repetitions.

The MV treatment of Japanese knotweed plants was carried out in the morning hours when the air temperature was 22.4 °C, air humidity was 85–89%, and soil moisture, determined by the dryer method in the laboratory, was 88%.

Both before and after MV treatment, the temperature of the plants was measured with a FLIR E60 thermal imaging camera equipped with Wi-Fi (manufacturer: FLIR Systems, Inc., Wilsonville, OR, USA), which measures the temperature in the range from −20 to +120 °C (±2 °C). The camera was placed approximately 0.5–1.0 m above the plant stems, and images were taken with a resolution of 320 × 240 pixels. Each image was then analyzed with FLIR ResearchIR MAX (manufacturer: Teledyne FLIR LLC, Wilsonville, OR, USA) based on mathematical processing of the pixel color scale—regression analysis in an Excel spreadsheet analyzed average temperatures for each MV time.

The efficiency of radiation control was checked at 10 and 56 days after treatment (DAT). The number of regrowing plants was counted for each MV treatment and replication. Moreover, all MV areas were photographed to document the outcomes.

### 4.3. Ecotoxicological Analyses of Soil Treated with Microwaves

Soil samples were collected on 29 June 2022, immediately after the plots with plants were MV-treated. Nine samples were taken from each MV plot and mixed into one collective sample. In total, six collective soil samples and a control sample were prepared. The soil was stored at 3–4 °C until further analysis. Soil ecotoxicity was tested using three biotests (Table 5). Two model plant species were used in the Phytotoxkit test: the monocotyledonous *Sorghum saccharatum* (L.) Moench and the dicotyledonous *Sinapis alba* L. After a 3-day incubation of the samples, two parameters were tested: germination inhibition (%) and root growth inhibition (%) [44]. In the Ostracodtoxkit test, the test organism was the crustacean *Heterocypris incongruens* (Ramdohr 1808) [45]. For this assay, after 6 days of incubation of the samples, mortality and inhibition of *H. incongruens* growth were measured. The Microtox system uses the luminescent bacteria *Aliivibrio fischeri* (Beijerinck 1889) [46]. Exposure of the bacteria to the sample disrupts the metabolic process, reducing the light produced by the bacteria. Measuring the inhibition of bacterial glow in the test sample relative to the control sample is a measure of toxicity evaluation. The toxicity of the soil samples to *A. fischeri* was tested using the 81.9% screening test using the M 500 Analyzer [47]. The aqueous extract was prepared by mixing 1 volume of the sample with 10 volumes of redistilled water and shaking it mechanically for 24 h. Luminescence was measured before and after a 15 min incubation of the bacterial suspension with the test sample. Screening tests were performed based on the analysis of undiluted samples. The samples’ toxicity was assessed based on the percentage value of the toxic effect (PE%) estimated for the performed biotests. The samples were assigned to the appropriate toxicity classes: class I (PE ≤ 20% no significant toxic effect)—non-toxic sample; class II (20% < PE ≤ 50% significant toxic effect)—low-toxic sample; class III (50% < PE < 100% significant toxic effect) toxic sample; class IV (PE = 100%) very toxic sample [47].

### 4.4. Analysis of Resveratrol Content in Rhizomes after Microwave Radiation

Healthy Japanese knotweed rhizomes obtained on 23 August 2022 from the field population were carefully washed under running water and divided into fragments of various lengths (1.7–20.9 cm) and diameters (1.5–3.2 cm) so that their total weight was 500 ± 2 g. The rhizomes were MV-treated in the laboratory using one antenna emitting MV of the surface power density of 30 kW/m^2^ for 2.5, 5.0, 7.5, or 10.0 min. The device was described in detail by Słowiński et al. [28]. Directly after MV, the temperature of the rhizomes was measured with the FLIR E60 thermal imaging camera (FLIR Systems, Inc., Wilsonville, Oregon, USA) and weighed at laboratory balance (Radwag, Radom, Poland, PL) with ±0.01 g accuracy. Next, the bark from the rhizomes was removed, and small (0.5–1.5 cm long) and thin (0.5–1.2 mm) slices were cut with a sterile razor into sterile containers. The material was lyophilized at −40 °C in Labconco FreeZone (Kansas City, MO, USA) freeze-dry system for ca. 30 h and sent to analyses for resveratrol content.

The analytical studies were performed using an Agilent Technologies 1260 Infinity ll (Santa Clara, CA, USA) series coupled with Agilent Technologie Infinitylab LC/MSD equipped with an electrospray ionization (ESI) source operating in negative ionization modes. The separation of the target compound was achieved on a Synergi Polar-RP C18 analytical column (250 mm × 4.6 mm, 4 μm) from Phenomenex (Chesire, UK). The mobile phase was a mixture of (A) water and (B) acetonitrile, with 0.1% formic acid, at a flow rate of 0.8 mL/min in gradient elution mode. The composition of the mobile phase varied as follows: starting with 20% B, then 0–10 min, 70% B; 10–15 min, isocratic condition, 70% B; 15–18 min, 20% B; after which, the column was reconditioned for 10 min. All solvents and solutions were filtered through a 0.2 μm polyamide filter from Sartorius Stedim (Goettingen, Germany). The injection volume was 5 μL. The column temperature was 30 °C, and the drying gas temperature in the ionization source was 350 °C. The gas flow was 12 L/min, the nebulizer pressure was 55 psi, the capillary voltage was 3000 V, and the fragmentor was 70. Detection was performed in single ion monitoring (SIM) mode. The extracted ion chromatogram of the ion current of the peak *m/z* 227, corresponding to the (M-H)-, was used to quantify resveratrol.

#### 4.4.1. Reagents and Standard

The standard of trans-resveratrol was purchased from Vinci-Biochem (Florence, Italy). At a concentration of 1000 mg/L, individual stock solutions of trans-resveratrol were prepared by dissolving the pure standard compound in LC-MS grade methanol and storing it in glass-stoppered bottles at 4 °C. Standard working solutions at various concentrations, from 0.25 to 2 mg/L, were prepared daily by appropriately diluting the stock solutions with HPLC-grade methanol. LC-MS-grade methanol and acetonitrile were purchased from Merck, while HPLC-grade formic acid 99–100% was bought from JT Baker BV (Deventer, Holland). Deionized water (>18 MΩ cm resistivity) was further purified using a Milli-Q SP Reagent Water System (Millipore, Bedford, MA, USA). Before HPLC analysis, all samples were filtered with Phenex™ RC 4 mm 0.2 μm syringeless filter, Phenomenex (Castel Maggiore, BO, Italy).

#### 4.4.2. Sample Preparation and Extraction Procedures

Lyophilized R. japonica samples were ground with a cutter mill using a 6 mm grid. Afterward, each sample was extracted with ethanol: water mixture (80:20, *v/v*) in a 1:10 drug–solvent ratio by sonication (FALC ultrasonic bath, Treviglio, Italy) at 25 °C and frequency of 40 KHz for 30 min. After centrifugation at 5000 rpm for 10 min with a Thermo Scientific IEC CL10 Centrifuge from Thermo Electron Industries SAS (Chateau-Gontier, France), the sample solutions were filtered through a 0.2 μm syringeless filter before injection into HPLC-MS/MS triple quadrupole. All samples were stored at 15 °C until analysis. Each sample was analyzed in duplicate.

#### 4.4.3. Method Validation

The HPLC-DAD-ESI-MS analytical method was validated by assessing the linearity, limit of detection (LOD), and limit of quantification (LOQ). Linearity was tested by injecting external standards at different concentrations and then plotting and calculating calibration curves with the respective determination coefficients (R^2^). Calibration curves were prepared using the concentration range of 0.25–2 mg/L. Resveratrol showed good linearity as the R^2^ was higher than 0.9995. LOD and LOQ were obtained by injecting serial dilutions of the corresponding standard solutions and calculating the signal-to-noise ratio (SNR). LOD, the smallest analyte detectable in the sample, was calculated as the standard concentration with SNR = 3. In contrast, LOQ is the smallest quantity of analyte that can be quantified and corresponds to the analyte amount for which the signal-to-noise ratio is equal to 10. The LOD for resveratrol was 100 ppb while LOQ was 333 ppb. Appendix A reports the extracted ion chromatogram (EIC) used for the analyte quantification.

## 5. Conclusions

The research showed that it is reasonable to use MV for at least 20–25 min to control Japanese knotweed. Applying MV to Japanese knotweed for shorter times is not effective, as the plants regenerate from underground rhizomes within a period of up to one month. MV does not negatively affect the ecotoxicity of the tested soil. The use of MV significantly reduces the content of resveratrol in knotweed rhizomes. Given the above results, we conclude that the microwave control of Japanese knotweed could be considered in the future as one of the non-chemical control methods, especially in the context of the Green Deal, (Farm to Fork strategy of the European Commission); however, further studies are needed.

## Figures and Tables

**Figure 1 plants-13-00152-f001:**
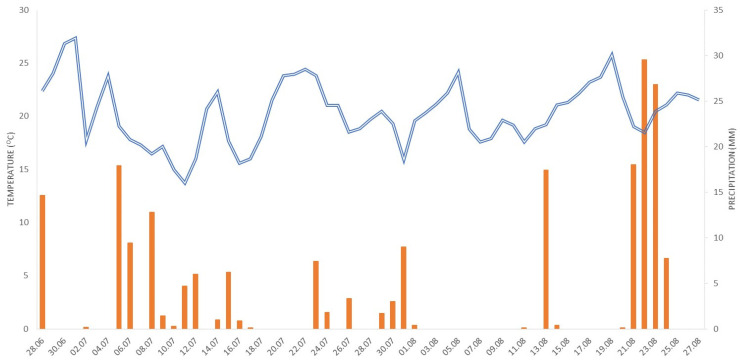
The average daily air temperatures in C (blue line) and daily precipitation sum in mm (orange bars) from 28 June to 27 August 2023 from the meteorological station in Balice.

**Figure 2 plants-13-00152-f002:**
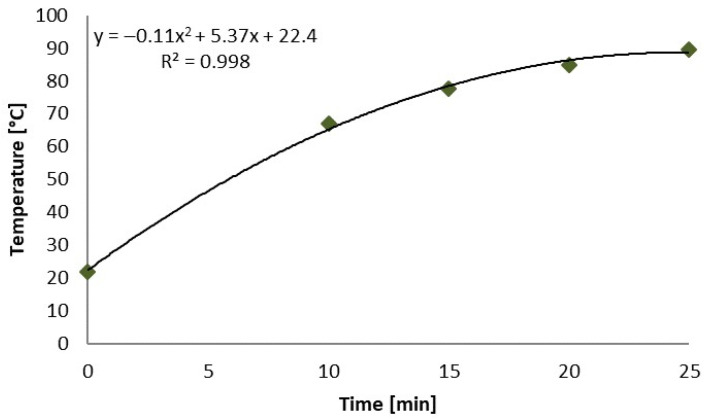
The curve and regression equation for the temperature (recorded by the FLIR E60 thermal imaging camera) of cut knotweed shoots and surrounding soil following radiation with microwaves for 5.0, 7.5, 10.0, 15.0, 20.0, and 25.0 min in field conditions.

**Figure 3 plants-13-00152-f003:**
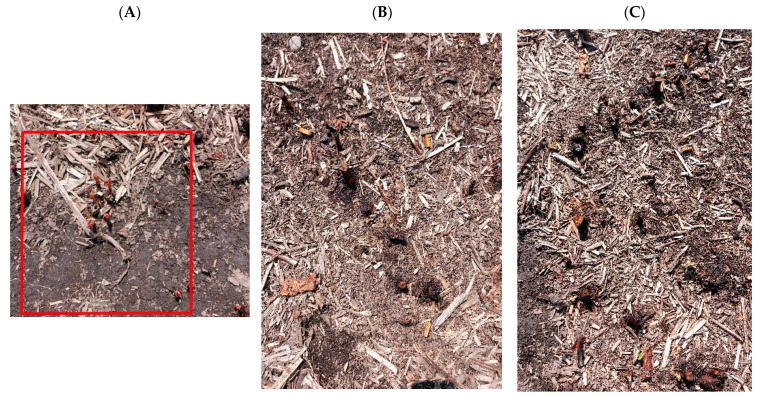
Japanese knotweed at 10 days after microwave radiation. Radiation times: (**A**) 5.0 min, (**B**) 20.0 min, (**C**) 25.0 min. In Figure 3A, the red square indicates new, regrowing Japanese knotweed shoots.

**Figure 4 plants-13-00152-f004:**
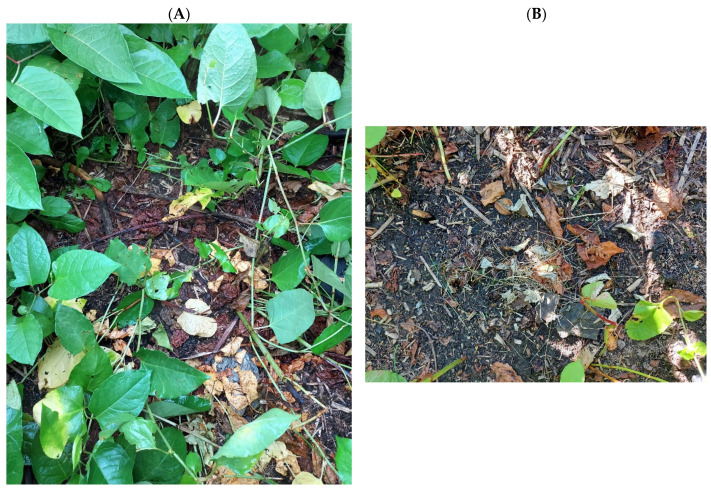
Japanese knotweed at 56 days after microwave radiation. Radiation times: (**A**) 20.0 min, (**B**) 25.0 min. Empty spots indicate absence of new shoots.

**Figure 5 plants-13-00152-f005:**
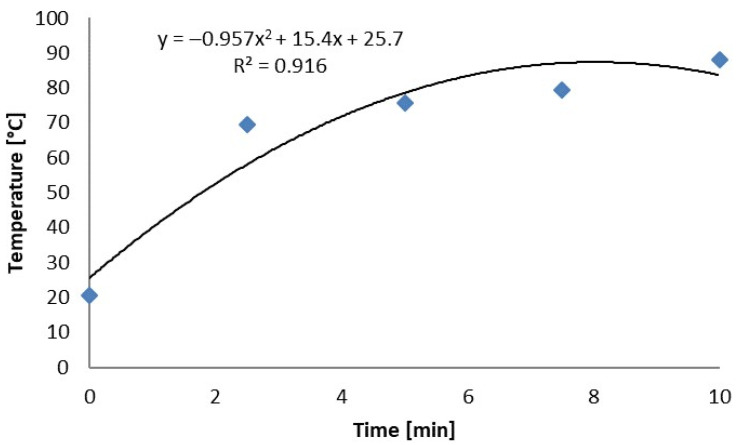
Regression curve for an air temperature of knotweed rhizomes microwaved in ambient conditions for 2.5, 5.0, 7.5, and 10.0 min.

**Figure 6 plants-13-00152-f006:**
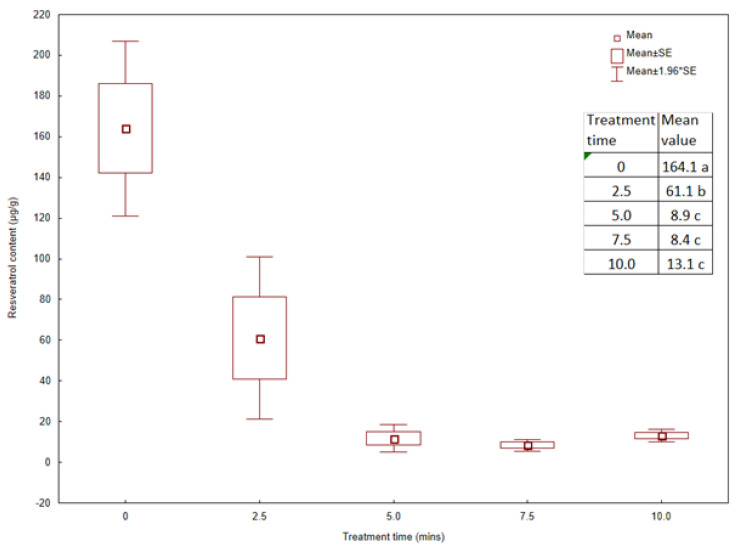
Resveratrol content in *Reynoutria japonica* extracts (µg/g) by HPLC-DAD-ESI-MS (*n* = 6). Treatment time refers to the time length of microwave radiation in min. Means were compared by one-way ANOVA, F = 24.3; *p* < 0.001. Different letters by the mean values in the table refer to significant differences according to the LSD test.

**Figure 7 plants-13-00152-f007:**
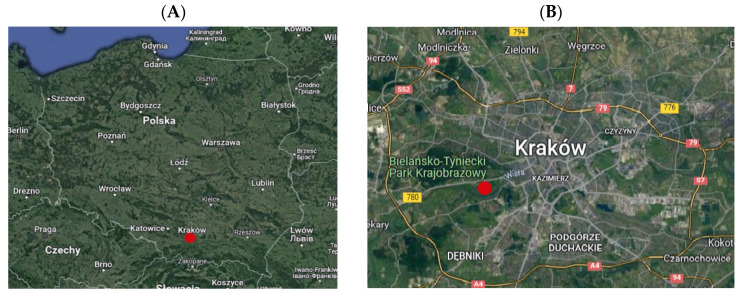
Location of study site in Poland (**A**) and Krakow (**B**).

**Figure 8 plants-13-00152-f008:**
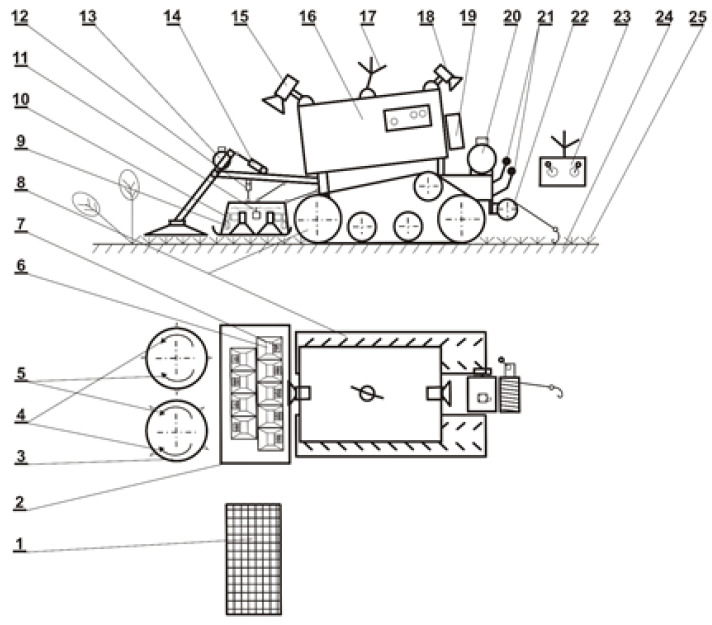
Scheme of a self-propelled HOGWEED device for microwave destruction of invasive plants. 1—air filter; 2—microwave emitter; 3—rotary mower; 4,5—direction of rotation depending on the plant development phase; 6—microwave antenna; 7—magnetron; 8—tracking system; 9—capacitor; 10—transformer; 11—cooling fan; 12—microwave emitter actuator; 13—mower drive motor; 14—mower actuator; 15—camera 1; 16—power generator; 17—remote control antenna; 18—camera 2; 19—control system; 20—drive electric motor; 21—gear levers; 22—cable winch; 23—remote control system; 24—ground; 25—plants.

**Table 1 plants-13-00152-t001:** The temperature analysis results of the thermograms (FLIR E60 thermal imaging camera) from the knotweed and soil treated with microwaves at times 0–25.0 min in a field experiment. The camera measurements were taken directly after the microwave radiation.

Radiation Time (min)	0.0	5.0	7.5	10.0	15.0	20.0	25.0
Mean (°C)	21.9	45.4	53.7	67.0	77.6	84.8	89.6
Standard Deviation (°C)	0.6	5.2	4.5	8.2	11.6	13.5	18.0
Maximum (°C)	24.3	56.1	64.0	95.9	112.3	137.7	150.2
Minimum (°C)	20.3	29.9	34.0	42.9	42.4	36.3	39.5

**Table 2 plants-13-00152-t002:** The effect of microwave radiation on soil ecotoxicity. ^1^ IG—inhibition of germination; ^2^ IR—inhibition of root growth; ^3^ M—mortality; ^4^ GI—inhibition of growth; ^5^ IL—inhibition of luminescence. ^6^ I—first toxicity class, II—second toxicity class. ^7^ mean values with different letters denote significant differences at *p* < 0.05.

Radiation	*S. alba*		*S. saccharatum*		*H. incongruens*		*A. fischeri*	Class Toxicity ^6^
	IG ^1^	IR ^2^	IG	IR	M ^3^	GI ^4^	IL ^5^	
	Percentage Toxic Effect PE%							
0 min (control)	0	42	0	5	0	41	9 b ^7^	II
5 min	-	-	-	-	0	13	−9 a	I
7.5 min	21	25	8	−14	0	35	−4 a	II
10 min	7	−10	0	−25	0	37	−3 a	II
15 min	14	21	0	−5	0	35	20 c	II
20 min	21	43	0	−7	0	37	22 c	II
25 min	14	37	0	−14	0	48	26 c	II

**Table 3 plants-13-00152-t003:** In the laboratory experiment, the temperature analysis results from the thermograms of the bare knotweed rhizomes treated with microwaves at times 0–10.0 min. The measurements were taken directly after the radiation.

Time of Radiation (min)	0.0	2.5	5.0	7.5	10.0
Mean (°C)	20.3	69.4	75.8	79.5	87.9
Std. Dev. (°C)	0.1	13.0	7.4	10.2	10.6
Maximum (°C)	20.8	101.7	90.1	108.8	116.0
Minimum (°C)	19.9	40.2	49.1	42.8	45.8

**Table 4 plants-13-00152-t004:** Characteristics of soil in the study site.

Sand %	Loam %	Clay %	pH -	C org. %	N g/kg	S g/kg	P g/kg	K g/kg	Ca g/kg
70	26	4	6.06	2.55	1.95	0.38	0.68	2.47	6.21
Mg g/kg	Na g/kg	Fe g/kg	Mn g/kg	Zn g/kg	Cu mg/kg	Ni mg/kg	Cd mg/kg	Pb mg/kg	Cr mg/kg
3.20	0.19	13.23	0.68	153.55	17.35	34.44	0.66	32.20	32.60

**Table 5 plants-13-00152-t005:** Characteristics of the ecotoxicological biotests.

Test	Model Organism	Final Parameter	Time
Phytotoxkit	*Sinapis alba*, *Sorghum saccharatum*	Inhibition of germination, inhibition of root elongation	72 h
Ostracodtoxkit	*Heterocypris incongruens*	Death, growth inhibition	6 dni
Microtox	*Aliivibrio fischeri*	Luminescence inhibition	15 min

## Data Availability

Data are available from the corresponding author on request.

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
