# Peer review of "Microwave Control of Reynoutria japonica Houtt., Including Ecotoxicological Aspects and the Resveratrol Content in Rhizomes"

_plants, 2024, doi:10.3390/plants13020152_

Round 1
Reviewer 1 Report
Comments and Suggestions for Authors
In the present work, the authors investigated the effectiveness of electromagnetic microwaves (MV) in destroying Japanese knotweed using an original device HOGWEED (MV of 2450 MHz), the ecotoxic effect of the MV on the soil environment, and the resveratrol content in knotweed rhizomes after MV treatment in 2022 in southern Poland. Cut plants were MV treated at times: 5.0 - 25.0 min. The MV efficiency was checked 10 and 56 days after treatment (DAT). After MV treatment, fresh soil samples were taken to analyze their ecotoxicity. As a result, at 56 DAT, knotweed was controlled if MV was used for at least 20.0 min. The MV did not affect the soil ecotoxicity. The MV-treated soils were classified as non-toxic or low-toxic. To analyze the resveratrol content, healthy knotweed rhizomes were dug out and treated with MV in the laboratory at times: 2.5 - 10.0 min, and analyzed for the resveratrol content in HPLC-MS/MS. As a result, the resveratrol in the rhizomes significantly decreased following MV exposure in a time-dependent manner. This work was well conducted and displays interesting results. Therefore, I suggest its acceptance after minor revision, as follows:
1. Please revise all scientific plant names and correct them to present in italics.
2. Figure 5 - temperature of cut knotweed shoots and surrounding soil following radiation with microwaves for 5.0, 7.5, 10.0, 15.0, 20.0, and 25.0 min in field conditions. Are these results obtained using one single point for each period? Why not in triplicate to give standard deviation? The same is true for the values presented in Figure 8.
3. Figure 4 indicated pure resveratrol, obtained from a standard compound. In my opinion, this figure must be withdrawn from the manuscript or transferred to supporting information (if available).
4. Please, the language must be carefully revised.
Comments on the Quality of English Language
The language must be carefully revised since some minor spelling were found in the text.
Author Response
Dear Reviewer, we would like to thank you for your comments. Below are step-by-step responses to your comments:
- Please revise all scientific plant names and correct them to present in italics.
All the scientific names of plants were revised and written using Italics.
- . Figure 5 - temperature of cut knotweed shoots and surrounding soil following radiation with microwaves for 5.0, 7.5, 10.0, 15.0, 20.0, and 25.0 min in field conditions. Are these results obtained using one single point for each period? Why not in triplicate to give standard deviation? The same is true for the values presented in Figure 8.
The thermal camera FLIR recorded the temperature, and Table 1 contains the results read from the camera, as the table title indicates. The same is true for the temperature values presented in Figure 8 (presently Figure 2).
3. Figure 4 indicated pure resveratrol, obtained from a standard compound. In my opinion, this figure must be withdrawn from the manuscript or transferred to supporting information (if available).
The Figure was moved to the Supplementary files.
4. Please, the language must be carefully revised.
The language was revised using the professional language editor Grammarly.
Reviewer 2 Report
Comments and Suggestions for Authors
Dear Authors,
The search for new methods of plant protection is an important element of modern agriculture. The withdrawal of active substances, legal regulations and social pressure to limit the application of chemical plant protection products contribute to the fact that this research direction is very important. The authors presented the results of their research in an accessible way, the continuation of which may contribute a lot to the development of new methods of weed control. In my opinion, however, it is worth adding some corrections to the text.
1. In keywords, add non-chemical weed control.
2. In the Conclusions, mention the use of this method in the context of the Green Deal.
Author Response
Dear Reviewer, we would like to thank you for your comments. Below please find step-by-step responses to your comments:
- In keywords, add non-chemical weed control.
Added
- In the Conclusions, mention the use of this method in the context of the Green Deal.
The Green Deal context is mentioned in lines 422-425.
Reviewer 3 Report
Comments and Suggestions for Authors
Dear authors,
you present an interesting alternative way to control invasive Reynoutria japonica. I recommend this article for publishing.
However, this research has limited use, but potential for more detailed study.
Author Response
You present an interesting alternative way to control invasive Reynoutria japonica. I recommend this article for publishing.
We sincerely thank the Reviewer for the positive comment above.
However, this research has limited use but potential for more detailed study.
The potential of the study is mentioned in Conclusions, line 425.